# Economic Evaluation of Artificially Intelligent (AI) Diagnostic Systems: Cost Consequence Analysis of Clinician-Friendly Interpretable Computer-Aided Diagnosis (ICADX) Tested in Cardiology, Obstetrics, and Gastroenterology, from the HosmartAI Horizon 2020 Project

**DOI:** 10.3390/healthcare13141661

**Published:** 2025-07-10

**Authors:** Magda Chatzikou, Dimitra Latsou, Georgios Apostolidis, Antonios Billis, Vasileios Charisis, Emmanouil S. Rigas, Panagiotis D. Bamidis, Leontios Hadjileontiadis

**Affiliations:** 1Pharmecons Easy Access Ltd., York YO31 0AA, UK; 2Signal Processing & Biomedical Technology Unit, Department of Electrical & Computer Engineering, Aristotle University of Thessaloniki, 541 24 Thessaloniki, Greece; gapostol@auth.gr (G.A.); vcharisis@ee.auth.gr (V.C.); leontios@auth.gr (L.H.); 3Lab of Medical Physics and Digital Innovation, School of Medicine, Aristotle University of Thessaloniki, 541 24 Thessaloniki, Greece; ampillis@med.auth.gr (A.B.); erigas@auth.gr (E.S.R.); bamidis@auth.gr (P.D.B.)

**Keywords:** economic evaluation, cost consequence analysis (CCA), digital health interventions (DHIs), coronary computed tomography angiography (CCTA), echocardiography, obstetrics, capsule endoscopy (CE)

## Abstract

**Objectives**: This study evaluates the economic impact of digital health interventions (DHIs) developed under the HosmartAI EU-funded program, focusing on obstetrics, cardiology, and gastroenterology. **Methods**: A Cost Consequence Analysis (CCA) was chosen in order to be able to examine the costs and consequences of AI technologies in early diagnosis of preterm births, echocardiography, coronary computed tomography angiography (CCTA), and capsule endoscopy (CE). **Results**: The results show that in obstetrics and CCTA, the AI technologies are cost-saving, with the AI-based preterm birth detection leading to savings of 99,840 EUR due to reduced severity of prematurity. In the echocardiography scenario, the new AI technology slightly increased costs (9409 vs. 2116 EUR), but offered benefits in diagnostic accuracy and shorter interpretation duration, particularly for less experienced physicians. Similarly, the capsule endoscopy AI technology raised annual costs by 6626 EUR but improved productivity, accuracy, and user satisfaction. **Conclusions**: The findings emphasize the need for standardized frameworks to guide economic evaluations of DHIs, ensuring informed healthcare investment and reimbursement decisions in the future.

## 1. Introduction

Diagnostic tests, including genetic and imaging tests, are essential health interventions used to detect the presence or severity of a disease [1,2]. Their development and introduction follow a process similar to that of other health technologies, such as therapeutic drugs. Likewise, diagnostic trials can be categorized based on different research phases, ranging from early exploratory studies to assessments of clinical impact [3]. Over the past 40 years, the field of diagnostic trials has experienced significant growth [4]. Artificial Intelligence (AI) is revolutionizing diagnostic imaging in healthcare by integrating advanced algorithms and machine learning to enhance the interpretation of medical images such as X-rays, MRIs, and CT scans. Beyond automating processes, AI fundamentally improves diagnostic accuracy and efficiency [5]. One of its key advantages is the ability to rapidly analyze medical images, significantly reducing diagnosis time compared to traditional methods, which can be slow and prone to human error. This speed is especially critical in emergency situations where timely decisions are essential. Additionally, AI enhances diagnostic precision by learning from vast datasets, allowing it to detect patterns and anomalies that might be missed by human observers [6,7]. This improved accuracy helps minimize misdiagnoses and ensures patients receive the appropriate treatment more quickly [7]. While accurately diagnosing a health condition is a crucial first step in patient management, medical decision-making ultimately depends on the overall net health benefit, including improvements in morbidity, mortality, functional status, and quality of life. Differences in diagnostic accuracy can influence treatment decisions, ultimately affecting disease prognosis and patient outcomes. Therefore, in the final development phase of new medical tests, it is essential to evaluate both the diagnostic process and its downstream impact on treatment decisions [8].

The four clinical trials of computer-aided diagnostic technologies from HosmartAI originated from type 1 diagnostic trials. While these trials were sufficient to establish diagnostic accuracy, further clinical evidence was incorporated, considering user experience outcomes to estimate the experience factor [9]. Beyond diagnostic accuracy, additional factors such as healthcare resource utilization and user usability were assessed. The goal was for the test–treatment trial to generate evidence of the efficiency of care by evaluating economic outcomes through Cost Consequence Analysis (CCA).

The study aimed to conduct an economic evaluation of AI-driven web-based predictive models in decision support systems across three therapeutic areas with four clinical scenarios: (a) obstetrics—early detection of preterm births, (b) cardiology—(i) echocardiography and (ii) coronary artery disease (CAD)—stenosis detection, and (c) gastroenterology—capsule endoscopy. Cost Consequence Analysis (CCA) is an economic evaluation approach that presents disaggregated costs alongside a range of outcomes, allowing decision-makers to assess their relevance based on their specific context [10]. CCA is particularly recommended for complex interventions with multiple effects and public health initiatives that generate both health and non-health benefits, which may be difficult to quantify using a single metric [11]. Unlike traditional health economic evaluations that focus on measures like quality-adjusted life-years (QALYs), CCA incorporates broader considerations, including patient-oriented outcomes and non-health-related factors. This approach is especially valuable to funders prioritizing patient-centered impacts and intervention costs. Additionally, CCAs can be useful in feasibility or pilot studies to help identify the most relevant costs and outcomes for future large-scale trials.

## 2. Methods

A micro-costing analysis was conducted to identify key cost components, including (a) development costs of the new AI technology, (b) maintenance and infrastructure costs, (c) diagnostic costs, (d) examination costs, (e) consumable costs, and (f) personnel costs (physicians and electrical engineers) [12]. Personnel cost data were sourced from budgetary control statements provided by the hospital’s finance department, covering salaries for full-time medical staff in each hospital department. Examination duration was recorded, and physicians’ time allocation was estimated based on their monthly salaries. Similarly, the cost of engineering personnel was calculated. Consumable costs were reported on a per-patient basis, with data obtained from the hospital’s supplies department. Operational and overhead costs were derived from budget control statements reflecting total hospital expenses. The cost of diagnostic tests and specialized medical examinations was reported on a per-case basis. Each technical solution was developed with dedicated funding of 50,000 EUR, which was allocated to support the design and development of the respective AI technology. For the purposes of economic evaluation, this amount was treated as a capital investment and depreciated over a 10-year period, resulting in an estimated annualized cost of 5000 EUR per solution. Infrastructure costs were based on the costs reported for each technology by the respective development team. In the case of capsule endoscopy and echocardiography scenarios, a GPU-equipped workstation was considered, whilst in the obstetrics and CCTA scenario virtual machine costs were applied. A 5-year discounted Cost Consequences Analysis for all technologies was performed to enable a more comprehensive decision-making process. A 3% discount rate was used in both costs and outcomes. Discounting started from year 2 onwards. One-way sensitivity analysis was performed, with a ±10% variation on the cost parameters to check the validity of the results.

The analysis was conducted from the perspective of the National Health System, with all costs expressed in 2024 Euros. Four prospective clinical trials were carried out, one for each tested technology, on a proof-of-concept basis, with trial results published elsewhere [13,14,15]. To assess the effectiveness of the new technology, selected Key Performance Indicators (KPIs) included (a) accuracy of detection, (b) interpretation time, (c) user satisfaction, and (d) for the obstetrics scenario, early detection of preterm births [16,17]. A Cost Consequence Analysis (CCA) was performed for each technology, allowing for the evaluation of multiple health and non-health-related outcomes. CCA considers a broad spectrum of costs and effects, presenting them separately to provide a comprehensive view of the intervention’s impact across different care sectors, in alignment with the impact inventory [17]. Incremental outcomes for the intervention and comparator were calculated across clinical, health economic, and overall economic measures. The economic incremental outcome was determined by summing the total costs incurred throughout the treatment duration for each strategy, with the comparator’s cost subtracted from the intervention’s cost [17]. The following equations were used:**Incremental Cost** = Mean cost of intervention (new AI technology) − Mean cost of comparator (current technology)**Incremental Effect (Consequences)** = Mean effect of intervention (new AI technology) − Mean effect of comparator (current technology)

Regarding the standard of care (current technology) of each therapeutic area specific protocols are followed which are described below. The standard of care in Greece for prenatal screening for preterm birth risk includes transvaginal ultrasound to measure cervical length (16–24 weeks) and clinical history. The prenatal screening protocols align with World Health Organization and local obstetric society’s guidelines. Hence, current practice represents non AI- clinical assessment, which is widely used in Greek hospitals. Diagnostic support tools based on machine learning or prediction algorithms are not routinely implemented. Tertiary hospitals have Neonatal Intensive Care Units (NICUs) and high-risk pregnancy units with better access to intervention, still regional or public hospitals may have limited ultrasound expertise and delays in identifying preterm birth especially in rural areas.

The standard of care for echocardiographic assessment follow guidelines of European Society of Cardiology (ESC). Left ventricular ejection fraction (LV-EF) and global longitudinal strain (LV-GLS) are measured manually by cardiologists or sonographers, depending on staff expertise. The use of automated AI tools for measurements is not yet standard. In routine practice experienced and less experienced physicians perform manual measurements of LV function.

Coronary CT Angiography is used for non-invasive coronary assessment, often combined with stress echocardiography or Fractional Flow Reserve (FFR) depending on European Society of Cardiology (ESC) guidelines. Interpretation is performed by experienced cardiologists without AI assistance in routine interpretation. The standard diagnostic pathway where CCTA is used alongside or as a follow-up to stress testing.

Capsule endoscopy is used mainly for suspected small-bowel bleeding, Crohn’s disease and small-bowel tumors particularly when colonoscopy and upper endoscopy are non-diagnostic [18]. As standard of care, a capsule endoscopy video is reviewed in two steps [19]. First, the entire video is quickly reviewed to identify areas of interest, a step called pre-reading. In some clinics, pre-reading is performed by experienced nurses or junior gastroenterologists (the pre-readers). After pre-reading, an expert gastroenterologist assesses the areas of interest to generate the final report. The current manual review process is time-consuming and labor-intensive. AI comparison aims to reduce the workload and reading time while maintaining diagnostic accuracy.

## 3. Results

Four separate Cost Consequence Analyses (CCAs) were conducted, each corresponding to a different technology. Each CCA incorporated all potential health outcomes and consequences, providing decision-makers with a comprehensive understanding of the intervention’s impact on both healthcare budgets and patient health.

**Early Diagnosis of Preterm Births:** The first analysis evaluated AI technology for the early detection of preterm births.

**Automated Estimation of LV-EF and LV-GLS in Echocardiography:** The second analysis focused on AI-driven automated measurements in echocardiography.

**Accurate Detection of Stenosis in Coronary Computed Tomography Angiography (CCTA):** The third analysis assessed AI technology for improved stenosis detection utilizing clinical information and imaging biomarkers extracted from CCTA.

**Automatic Detection and Classification in Capsule Endoscopy:** The fourth analysis examined AI’s role in automated detection and classification in capsule endoscopy.

Table 1 presents the four distinct medical scenarios.

Scenario 1: Cost Consequence Analysis of Preterm Births Early Diagnosis

The cost analysis indicates that the new AI-based technology is a cost-saving alternative compared to current clinical practice. While CADXpert OB-GYN incurs additional costs for personnel, maintenance, and AI infrastructure, its predictive capabilities enable closer monitoring of pregnant women, effectively delaying premature delivery by 3 to 4 weeks. This delay shifts the birth classification from prematurity with comorbidities (birth weight 1500–1999 g) with a Diagnosis-Related Group (DRG) cost of 3158 EUR to prematurity without comorbidities, which incurs a lower DRG cost of 2646 EUR, resulting in a cost reduction of 512 EUR per neonate. Considering the 195 preterm births of the study sample, this shift is estimated to generate total savings of 99,840 EUR. However, this estimate is conservative, as the actual cost savings could be significantly higher when factoring in the potential reduction in comorbidities and mortality associated with prematurity. The improvement in performance from 0.75 to 0.83 is statistically significant as well as the diagnosis timing by 6 weeks (from 28 to 34 weeks gestation). Both outcomes are clinically relevant since the earlier diagnosis allows timely intervention leading to reduction of neonatal morbidity and mortality. Table 2 presents the Cost Consequence Analysis of the obstetrics scenario.

**Costs**: The HosmartAI intervention incurs some additional upfront costs (AI technology, maintenance, and infrastructure), but it leads to significant savings through reduced costs for premature births (both with and without comorbidities). The overall savings amount to 99,840 EUR annually. The analysis was based on the 195 premature births corresponding to 207 pregnant women included in the study, representing the proportion of successfully identified cases through the AI-enabled early diagnosis system. According to the relevant Diagnosis-Related Group (DRG) coding, the cost of hospitalization for premature neonates with a birth weight of 1599–2000 g and comorbidities (DRG T25Mγ) amounts to 3158 EUR per infant. For neonates within the same birth weight range but without comorbidities (DRG T25X), the corresponding cost is 2646 EUR. It was conservatively assumed that, due to the AI-enabled delay in preterm delivery by approximately 4 to 6 weeks, neonates would reach at least 34 weeks of gestation and would likely avoid comorbidities. Therefore, the cost reduction of 512 EUR per case was applied across the study sample. **Outcomes**: The intervention improves clinical performance and has a longer diagnostic duration, indicating it may provide a more thorough diagnostic approach, but with a trade-off of more time required.

In conclusion, while the AI intervention increases certain costs, it leads to potential substantial savings in healthcare costs related to premature births and offers improved clinical outcomes. In the discounted 5-year horizon analysis the HosmartAI intervention shows a clear economic and clinical benefit by providing substantial cost savings from fewer and less severe premature births. Consistent clinical improvements in diagnostic accuracy and quick offset of upfront costs by high impact outcome improvement.

2.Scenario 2: Cost Consequence Analysis of Echocardiography

The annual cost of the new echocardiography technology is higher than the current practice (9409 vs. 2116 EUR), primarily due to the introduction of the AI-based system. Despite the higher costs, the HosmartAI technology reduces the examination reading time by 2.5 min (as shown in Table 2). Although this time savings results in fewer working hours required by cardiologists, the relatively low salaries of physicians are insufficient to offset the additional costs associated with the new technology. An important advantage of the new AI technology is its ability to allow junior cardiologists to perform echocardiography reviews more quickly and accurately. Furthermore, the clinical precision of the AI system surpasses that of the current technology, and user satisfaction remains at an acceptable level of 75%. LV-EF and LV-GLS measurements guide heart failure diagnosis, management, and prognosis. Improvements in diagnostic accuracy (Youden’s J from 0.54 to 0.80 for low-experience physicians, +0.26 is statistically significant), which might translate into better detection of cardiac dysfunction, reducing misdiagnosis and inappropriate treatment, still Youden’s J increase of 0.26 likely reflects a meaningful clinical improvement. Table 3 presents the Cost Consequence Analysis of the echocardiography scenario.

**Costs**: The HosmartAI intervention is more expensive overall due to personnel, maintenance, and infrastructure costs. However, it leads to savings on physician costs due to greater efficiency.

**Outcomes**: The intervention improves clinical performance by increasing diagnostic accuracy and reducing the time required for measurements, especially for less experienced physicians. It also improves system usability and user satisfaction and offers a higher level of accuracy and efficiency compared to current practice.

In the discounted 5-year horizon analysis there is a substantial gain in diagnostic accuracy for both less and more experienced physicians. Although the time efficiency gains are modest in absolute time (minutes), may be significant at scale based on the number of patients per year. The savings from reduced physician time for measurements do not fully offset the AI system costs. The cost of developing the technology and the maintenance costs, although not high, appear to exceed those of current practice. However, this is due to the particularly low salaries of physicians. If the same analysis was conducted in another European country with higher medical salaries, then the cost of the current practice would be higher, and the new technology would be cost-saving. The AI-supported echocardiography workflow saves 108 h of physician time annually, which is worth approximately 6480 EUR in staff cost alone. When combined with improvements in diagnostic accuracy and clinical workflow, the time savings strongly support efficiency, offering operational benefit of the technology by reducing human resources.

3.Scenario 3. Cost Consequence Analysis of CCTA Scenario

The cost analysis shows that the HosmartAI technology for CCTA examinations is a more cost-effective option compared to the current practice. The annual cost for all patients (n = 239) using HosmartAI is estimated at 41,860 EUR, significantly lower than the 67,210 EUR required for the existing system. The AI technology’s higher sensitivity reduces the need for CCTA exams, as it identified that 194 out of 286 patients did not have stenosis greater than 50%, resulting in 159 unnecessary CCTA exams being avoided. Moreover, the diagnosis process is quicker with the AI system, enabling less experienced physicians to interpret results faster and with greater accuracy. As a result, the budget impact savings from adopting the new AI-powered CCTA technology are estimated at 25,350 EUR annually for the hospital department. Diagnostic accuracy increased from 77% to 84%, with Youden’s J having a 0.07 absolute rise are both statistically significant results. The 7% increase in diagnostic accuracy is within the clinically important range. From economic perspective, the one fewer examination to diagnose per patient is important since it implies less radiation, reduced waiting time and costs. Table 4 presents the Cost Consequence Analysis of the CCTA scenario.

**Costs:** The annual cost of using the HosmartAI technology is 41,860 EUR, which is significantly lower than the 67,210 EUR required for the current system, resulting in 25,350 EUR in savings. Regarding examination costs, the AI technology reduces the cost of each CCTA examination by 159 EUR, and the overall cost of CCTA exams for 159 patients annually is reduced by 37,206 EUR.

**Outcomes:** The new technology shows improved clinical performance, with a score of 84.00% compared to the current practice’s 77.00%, reflecting a 7%. The duration of diagnosis is also reduced by 1 examination for both experienced and less experienced physicians. Experienced physicians can now complete 3 examinations with the HosmartAI system, as opposed to only 2 with the current practice. Similarly, less experienced physicians also benefit from the new technology, completing 3 examinations instead of 2.

The HosmartAI implementation in the CCTA setting delivers both cost savings and clinical improvements over a 5-year period. There is an economic benefit from the lower exam cost per patient as well as an improved clinical benefit by improved performance reduction of imaging resources. The fact that 159 CCTA examinations are avoided annually representing approximately 67% of the 239 patients, was achieved through the application of AI algorithms to assign risk scores in several parameters (i.e., Age, sex, chest pain, risk factors) and pre-test probability models to identify patients with a very low likelihood of disease. When the AI indicates a low probability of CAD, patients may safely avoid CCTA without compromising diagnostic accuracy.

4.Scenario 4. Cost Consequence Analysis of Capsule Endoscopy Scenario

The annual cost of the new AI-powered capsule endoscopy technology is slightly higher than the current practice (68,300 vs. 61,674 EUR), resulting in an additional annual cost of 6626 EUR. This increase is primarily due to the implementation of the new AI technology and its associated infrastructure. Despite the higher costs, the new technology offers time-saving benefits in terms of productivity. Particularly, it reduces the need for pre-reading, i.e., a preliminary video screening typically performed by specialized nurses or junior gastroenterologists.

Thus, it shortens review time and requires fewer personnel, with only one healthcare professional needed for analysis instead of two. The review process is also significantly faster, taking only 60 min with the AI technology versus 240 min with the traditional method. In terms of effectiveness, the AI technology performs similarly to the current practice, with sensitivity rates of 0.89 versus 0.90. However, the time savings and reduced labor requirements are not sufficient to offset the additional cost when compared to the current practice. User satisfaction with the new technology is notably high, reaching 76.4%. The reading time per case is reduced from 240 min to 60 min (−180 min) which is statistically significant. The removal of a junior doctor from the workflow is also economically important leading to economic efficiency. The 75% reduction in reading time (−3×) is clinically meaningful. Overall, while the HosmartAI intervention incurs higher costs, it offers improved clinical performance, reduces time per diagnosis, and increases productivity by requiring fewer physicians for the review process. Table 5 presents the Cost Consequence Analysis of the capsule endoscopy scenario.

**Costs**: The total annual cost of the HosmartAI intervention is 68,300 EUR, which is 6626 EUR higher than the current practice (61,674 EUR). The additional costs for the new AI technology are mainly attributed to personnel (5000 EUR), maintenance (3000 EUR), and infrastructure (300 EUR). However, the cost of the capsule itself remains the same in both scenarios (60,000 EUR). Additionally, the new technology eliminates the need for small bowel capsule endoscopy reviews, saving 1674 EUR.

**Consequences**: The AI-assisted capsule endoscopy improves system usability (76.4% vs. 70%) while maintaining similar diagnostic sensitivity (0.89 vs. 0.90). It significantly reduces the reading time from 240 to 60 min, enhancing efficiency. Additionally, only one senior doctor is required for screening, eliminating the need for a junior doctor, optimizing resource allocation

While HosmartAI in capsule endoscopy incurs a modest cost increase over 5 years, it delivers substantial workflow efficiency benefits including significantly reduced reading time and less human resources.

### Sensitivity Analysis

A sensitivity analysis was performed with a ±10% variation on the cost parameters to check the validity of the results. Most parameters show only modest influence when varied by ±10%. In the case of prematurity, the cost of averted severity of prematurity is the most influential parameter in the analysis. With +10%, the annual savings range from 89,856 to 109,824 EUR. This wide range reflects a strong impact on the overall result, but importantly, in all scenarios the intervention remains cost-saving.

In the context of Echocardiography, even with +10% variation, the intervention remains more costly than current practice, with the total difference ranging from 6563 to 8021 EUR annually. This indicates that the cost increase due to AI technology and maintenance is not offset by physician cost savings under current assumptions.

At CCTA scenario, the main driver of savings is the lower cost per CCTA examination under HosmartAI (base cost −159, range −143 to −175), especially when applied across a large patient volume. AI-related costs (personnel, maintenance) are not large enough to outweigh these savings. Even at sensitivity analysis, the intervention continues to generate annual cost savings.

Sensitivity analysis for the Capsule Endoscopy technology ranging from 5963 to 7289 EUR compared to current practice and the intervention remains more expensive in all scenarios. Savings from reduced review workload partially offset the AI-related costs but not completely.

Overall, the results of the analysis are robust to reasonable variations in individual cost estimates.

## 4. Discussion

The economic evaluation of Artificial Intelligence (AI) technologies in computer-aided diagnostic systems is essential for assessing their value within healthcare systems. The Cost Consequence Analysis (CCA) of the Interpretable Computer-Aided Diagnosis (ICADX) suite of software applications, developed under the HosmartAI initiative, provides valuable insights into the economic impact of AI technologies across multiple medical fields, including cardiology, obstetrics, and gastroenterology. In the obstetrics and CCTA scenarios, the new AI technologies were found to be cost-saving. These findings suggest that AI can enhance the effectiveness and efficiency of diagnostic processes, reducing overall costs. However, in the echocardiography and capsule endoscopy scenarios, the new technologies were slightly more expensive than the current methods. Despite this, the cost differences can be justified over time by the reduced duration required for interpretation and increased diagnostic accuracy. This highlights the longer-term benefits of AI, where the initial investment is offset by the improved efficiency and accuracy it offers, leading to overall cost savings. Our findings align with those of Bharadwaj P. et al. (2024) [20], who evaluated an AI-powered radiology diagnostic imaging platform and demonstrated a significant return on investment (ROI). Their study highlighted that AI integration reduced labor time, specifically in triage, reading, and reporting, while simultaneously enhancing patient outcomes [20]. Our study also emphasizes the need for the continuous training and adaptation of clinicians to maximize the full potential of AI systems, which is also a similar finding of the international literature [21]. In the obstetric context, AI’s role in improving the detection and monitoring of preterm births has shown great promise. While there is a scarcity of specific economic evaluations in this domain, AI’s potential to streamline workflows and enhance diagnostic precision suggests a favorable economic impact [22]. The ability of AI to reduce diagnostic errors and enhance patient management could lead to long-term savings and better outcomes. In gastroenterology, the current CCA of AI-assisted video capsule endoscopy is the only economic evaluation available in literature, although there is available AI research in the field [23]. Though a cost-effectiveness study comparing video capsule endoscopy with traditional strategies for managing acute upper gastrointestinal hemorrhage in the emergency department showed favorable results, it did not focus on the impact of digital technology specifically. This highlights the need for further studies to evaluate the full economic benefits of AI in gastroenterology, especially in comparison with traditional diagnostic methods [24].

In cardiology, the integration of AI into echocardiography and CCTA has been shown to enhance diagnostic accuracy and efficiency with a minimal budget impact. These results are consistent with findings from a systematic review examining the cost-effectiveness of digital health interventions (DHIs) in managing cardiovascular diseases (CVDs). The review, which included 14 studies, found that DHIs, such as telemonitoring, video conferencing, and mobile apps, resulted in higher quality-adjusted life-years (QALYs) and significant cost savings. These studies suggest that digital interventions are cost-effective in managing CVDs, and AI technologies in cardiology are showing similar promise in improving diagnostic outcomes and managing healthcare costs [25]. Another systematic review of health economic evaluations for AI-based health interventions included 21 studies, most of which focused on automated image analysis for screening and diagnosis, particularly in general medicine and oncology. This further supports the growing body of evidence indicating that AI applications, particularly in imaging, are valuable tools for improving diagnostic accuracy while maintaining or reducing healthcare costs [26]. The findings from this study, alongside the previous literature, demonstrate that while the initial costs of AI technologies may be higher, their long-term benefits, such as increased diagnostic accuracy, efficiency, and cost savings, outweigh the initial investment. Another systematic review, which analyzed 200 studies, revealed that AI is more cost-effective than traditional methods, and that AI applications in treatment yield greater economic benefits than in diagnosis. The findings highlight significant cost savings from AI implementation and emphasize that future gains can be enhanced through strategies such as pruning, reducing algorithmic bias, improving explainability, and achieving regulatory approval [27]. A recent study by Wang et al. (2024) [28] in a diabetic retinopathy screening program performed in China, comprising 251,535 participants with diabetes, demonstrated that regarding high diagnostic accuracy in diabetic retinopathy (DR) screening, the most accurate model was not always the most cost-effective. A large-scale cost-effectiveness analysis of that program in China revealed that only a subset of AI performance scenarios were either cost-saving or cost-effective, despite high sensitivity and specificity. Notably, the AI model with the highest accuracy (93.3% sensitivity, 87.7% specificity) was outperformed in economic terms by models with slightly lower specificity. These findings highlighted that improved accuracy can increase costs without proportionate benefits, emphasizing that AI implementation must be guided not only by technical performance but by independent economic evaluation to avoid non-cost-effective use [28]. The successful integration of AI in diagnostic protocols across various fields is promising and suggests a transformative potential for healthcare systems, provided that clinicians receive adequate training to adapt to these new technologies. While the capsule endoscopy scenario demonstrates a reduction in required personnel—specifically, one less junior doctor involved in the reading process—this efficiency gain could indeed introduce an opportunity cost in the form of reduced training exposure for junior clinicians. In traditional workflows, junior doctors play a critical role in reviewing capsule endoscopy videos, developing pattern recognition skills, and gaining experience in interpreting gastrointestinal abnormalities. By automating a large portion of this task, AI may limit hands-on training opportunities, potentially contributing to “de-skilling” over time. This concern is particularly relevant in specialties where diagnostic expertise is built gradually through repetitive exposure to varied and subtle pathology. As a mitigation risk, healthcare systems must balance automation with preserving clinical training programs. Failing to do so could impair long-term diagnostic capabilities and reduce workforce preparedness. Therefore, AI should augment and not replace medical education, especially in high-cognitive specialties.

Although AI technologies present valuable opportunities for diverse healthcare stakeholders, still their implementation should be tailored to specific system contexts. For public health systems, particularly in resource-constrained settings, these AI solutions can enhance diagnostic accuracy and reduce clinician workload—making them well-suited for addressing workforce shortages and improving access, especially in underserved regions. For private healthcare providers, the focus may lie in improving efficiency, patient throughput, and market competitiveness through time savings and workflow optimization. Policymakers and health technology assessment (HTA) bodies should consider phased adoption strategies that include early pilot studies, real-world evidence generation, and the continuous monitoring of clinical and economic impact. Importantly, different financing models (e.g., outcome based vs. fee-for-service) will influence cost-effectiveness thresholds, requiring context-specific HTA frameworks. Gradual integration, supported by training and digital infrastructure investment, will be essential for sustainable and scalable adoption across healthcare settings.

## 5. Limitations

While the preliminary results of HosmartAI technologies are promising, several limitations must be acknowledged. The studies were conducted in single-center or limited settings, which restricts the generalizability of findings across diverse healthcare environments. Most evaluations remain at the proof-of-concept stage, making it difficult to draw conclusions about real-world scalability or long-term clinical and economic impacts. Additionally, the short time horizon used in these assessments may overlook delayed benefits or downstream costs. The absence of patient-reported outcomes limits the ability to capture user-centered value, such as satisfaction or quality-of-life improvements. Finally, there is a potential selection bias, as participating physicians may have been early adopters or more technologically inclined, which could overestimate usability and acceptance in broader clinical practice. These factors highlight the need for larger, multi-center studies with extended follow-up and broader stakeholder inclusion. The current analysis utilized a Cost Consequence Analysis (CCA), which encourages the presentation of disaggregated measures across various health and non-health impacts in different care sectors, in line with the impact inventory. However, the absence of the traditional cost-per-QALY (quality-adjusted life-year) assessment, which is often used in NICE’s willingness-to-pay thresholds, means that decision-makers must rely on their own value judgments to determine whether the benefits of digital technologies justify the additional costs. For instance, some decision-makers may prioritize the improvements in patient well-being brought by the new technologies, while others may place greater importance on the enhanced efficiency of healthcare delivery [29]. Furthermore, several challenges remain that could impact the full implementation and integration of AI technologies in clinical settings. Additionally, the need for extensive validation studies is essential to ensure that the AI technologies perform consistently and safely across diverse patient populations and clinical environments, since the current studies were proof of concept. Lastly, meaningful patient and community involvement in the development of AI applications is vital to ensure that the technologies meet the needs and expectations of those they are intended to serve. These factors should be addressed in future studies and implementations to maximize the potential of AI in healthcare [30].

As the healthcare sector continues to embrace AI and other digital technologies, a comprehensive approach to evaluating their economic impact will become increasingly important. The development of such frameworks will provide healthcare decision-makers with the tools necessary to assess the value of DHIs holistically, considering both direct and indirect costs, as well as the broader implications for patient care and system efficiency. Future economic evaluations should consider not only the financial aspects of these technologies but also their long-term impact on healthcare delivery, patient quality of life, and health system sustainability [31].

## 6. Conclusions

The digital health interventions (DHIs) developed under the HosmartAI EU-funded program demonstrate significant promise in enhancing diagnostic accuracy, improving patient outcomes, and achieving high levels of user acceptance across various medical specialties, including obstetrics, cardiology, and gastroenterology. The technologies analyzed in obstetrics and CCTA scenarios reveal substantial cost savings, while the echocardiography and capsule endoscopy technologies, though slightly more expensive initially, demonstrate long-term value due to their time-saving benefits and improved diagnostic precision. In all cases, the AI interventions offer more efficient workflows and reduce diagnostic errors, which ultimately leads to better patient management and overall cost savings. Despite the growing adoption of DHIs, the integration of economic evaluations into their development remains limited. This analysis highlights the critical need for a standardized framework that can guide the generation, synthesis, and interpretation of economic evidence related to DHIs. Such a framework is essential not only to ensure that the economic benefits of these technologies are fully understood but also to inform critical decisions regarding reimbursement and investment. Given the inherent complexity of DHIs, including the diverse range of health and non-health outcomes they impact, this framework should help identify potential challenges, tailor appropriate evaluation methodologies, and account for uncertainties in the economic assessment process.

## Figures and Tables

**Table 1 healthcare-13-01661-t001:** Medical Scenarios Under Evaluation.

Scenario	Clinical Area	AI Function
1. Early diagnosis of preterm births	Obstetrics	Early risk detection of preterm birth using maternal clinical data
2. Automated estimation of LV-EF and LV-GLS	Cardiology (Echocardiography)	Automatic measurement of cardiac function metrics (LV-EF, LV-GLS)
3. AI-assisted detection of coronary stenosis (CCTA)	Cardiology (Imaging)	AI analysis of CCTA data and clinical biomarkers to detect stenosis
4. Automated capsule endoscopy interpretation	Gastroenterology	Detection and classification of small bowel abnormalities

**Table 2 healthcare-13-01661-t002:** Cost Consequence Analysis—Obstetrics Scenario.

Cost Consequence Analysis—Obstetrics Scenario				
Cost/Outcome Categories	HosmartAI Intervention (Annual Cost)	Current Practice (Annual Cost)	Difference	Discounted 5-Year Total (EUR)
Cost of AI technology (personnel)	5000 EUR	0 EUR	5000 EUR	23,585 EUR
Cost of maintenance	6972 EUR	0 EUR	6972 EUR	32,897 EUR
Cost of AI infrastructure	170 EUR	0 EUR	170 EUR	802 EUR
Cost of prematurity (age category 1500–1999 g) with co-morbidity (T25Μγ) per case	0 EUR	3158 EUR	−3158 EUR	−14,895 EUR
Cost of prematurity (age category 1500–1999 g) without comorbidity (DRGT25X)	2646 EUR	0 EUR	2646 EUR	12,481 EUR
Cost of averted severity prematurity (n = 195 preterm births annually) annual cost	515,970 EUR	615,810 EUR	−99,840 EUR	−470,955 EUR
**Consequence Categories**	**HosmartAI Intervention**	**Current Practice**	**Difference**	**Discounted 5-Year Total**
Clinical performance	0.83	0.75	0.08	0.377
Duration of diagnosis PTB > 75% (duration for experienced physician > 2 years) in weeks	34 weeks	28 weeks	6 weeks	28.3 weeks

**Table 3 healthcare-13-01661-t003:** Cost Consequence Analysis—Echocardiography Scenario.

Cost Consequence Analysis—Echocardiography Scenario				
Cost/Outcome Categories	HosmartAI Intervention (Annual Cost)	Current Practice (Annual Cost)	Difference	Discounted 5-Year Total (EUR)
Cost of AI technology (personnel)	5000 EUR	0 EUR	5000 EUR	23,585 EUR
Cost of maintenance	3000 EUR	0 EUR	3000 EUR	14,151 EUR
Cost of AI infrastructure	300 EUR	0 EUR	300 EUR	1415 EUR
Physician cost of LV-EF measurement (n = 2880 patients annually)	634 EUR	1382 EUR	−749 EUR	−3534 EUR
Physician cost of LV-GLS measurement (n = 1440 patients annually)	475 EUR	734 EUR	−259 EUR	−1222 EUR
Total cost per year	9409 EUR	2116 EUR	7292 EUR	34,395 EUR
**Consequence Categories**	**HosmartAI Intervention**	**Current Practice**	**Difference**	**Discounted 5-Year Total**
System usability (SUS)	75.00%	-	75.00%	Not time-dependent
Mean absolute error (MAE) of automatic measurement of LV-EF	5.55	-	5.55	Constant
Accuracy of the automated analysis	3.03	-	3.03	Constant
Diagnostic accuracy for a low-experience physician (low-experience physician: <5 years) (Youden’s J index)	0.80	0.54	0.26	1.23
Diagnostic accuracy for a highly experienced physician (highly experienced physician: >5 years) (Youden’s J index)	0.82	0.64	0.18	0.85
Average time for measurement of LV-EF by a highly experienced physician (in min)	1.00	2.00	−1.00	−4.72
Average time for measurement of LV-EF by a low-experience physician (in min)	1.00	1.00	0.00	-
Average time for measurement of LV-GLS by a low-experience physician (in min)	1.50	4.00	−2.50	−11.79
Average time for measurement of LV-GLS by a highly experienced physician (in min)	1.50	3.00	−1.50	−7.08

**Table 4 healthcare-13-01661-t004:** Cost Consequence Analysis—CCTA Scenario.

Cost Consequence Analysis—CCTA Scenario				
Cost/Outcome Categories	HosmartAI Intervention (Annual Cost)	Current Practice (Annual cost)	Difference	Discounted 5-Year Total (EUR)
Cost of AI technology (personnel)	5000 EUR	0 EUR	5000 EUR	23,585 EUR
Cost of maintenance	6972 EUR	0 EUR	6972 EUR	32,897 EUR
Cost of AI infrastructure	170 EUR	0 EUR	170 EUR	802 EUR
CCTA examination	127 EUR	286 EUR	−159 EUR	-
CCTA examination (n = 239 patients annually)	29,718 EUR	66,924 EUR	−37,206 EUR	−175,594 EUR
Total cost per year	41,860 EUR	67,210 EUR	−25,350 EUR	−118,310 EUR
**Consequence Categories**	**HosmartAI Intervention**	**Current Practice**	**Difference**	**Discounted 5-Year Total**
Clinical performance	84.00%	77.00%	7.00% (0.07)	0.33
Duration of diagnosis (duration for experienced physician > 2 years)	3 examinations	2 examinations	1 examination	−4.72 fewer exams
Duration of diagnosis (duration for experienced physician < 2 years)	3 examinations	2 examinations	1 examination	−4.72 fewer exams

**Table 5 healthcare-13-01661-t005:** Cost Consequence Analysis—Capsule Endoscopy.

Cost-Consequences Analysis—Capsule Endoscopy Scenario				
Cost/Outcomes Categories	HosmartAI Intervention (Annual Cost)	Current Practice (Annual Cost)	Difference	Discounted 5-Year Total (EUR)
Cost of AI technology (personnel)	5000 EUR	0 EUR	5000 EUR	23,585 EUR
Cost of maintenance	3000 EUR	0 EUR	3000 EUR	14,151 EUR
Cost of capsule	60,000 EUR	60,000 EUR	0 EUR	0 EUR
Small bowel capsule endoscopy review	0 EUR	1674 EUR	−1674 EUR	−7897 EUR
Cost of infrastructure	300 EUR	0 EUR	300 EUR	1415 EUR
Total cost per year	68,300 EUR	61,674 EUR	6626 EUR	31,254 EUR
**Consequence categories**	**HosmartAI Intervention**	**Current Practice**	**Difference**	**Discounted 5-year Total**
System usability (SUS)	76.40%	70%	6%	Not time-dependent
Sensitivity of automated detection of small bowel abnormalities	0.89	0.90	−0.01	Slight reduction
Average time for completion of small bowel VCE reading (in min)	60.00	240	−180.00	−849 min
Number of personnel involved in screening/examination	1 senior doctor	1 junior doctor and 1 senior doctor	junior doctor −1	Workload Impact (less resources)

## Data Availability

Data are contained within the article.

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
