# Peer review of "Economic Evaluation of Artificially Intelligent (AI) Diagnostic Systems: Cost Consequence Analysis of Clinician-Friendly Interpretable Computer-Aided Diagnosis (ICADX) Tested in Cardiology, Obstetrics, and Gastroenterology, from the HosmartAI Horizon 2020 Project"

_healthcare, 2025, doi:10.3390/healthcare13141661_

Round 1
Reviewer 1 Report
Comments and Suggestions for Authors
General observations:
The article was presented well and was informative, including to a non-medical and non-technical audience. The categorisations were clear and so was the framework of analysis and the high level takeaways. I recommend for the publication of this article with minor revisions.
Specific observations:
- The four clinical scenarios listed from lines 66-69 were unclear and could perhaps be presented in a different way, perhaps as a list.
- The CCA framework of analysis was presented in lines 72-77. However, what are the drawbacks of using the CCA approach? Perhaps this could be addressed. The use of CCA for these four clinical scenarios could also be justified
- The presentation of the 4 clinical scenarios under the Results section can be improved. Right now, I cannot easily tell the difference between one section of a clinical scenario and others. This relates to providing proper headings for lines 127, 151 for example
- When comparing new AI interventions with current technologies and practices, the current technologies used should be mentioned somewhere. Comparisons work better with clear baselines. This could be mentioned in scenarios 1 and 2.
- Some clarification is also needed for the cost of AI infrastructure. Is this a one-off cost or an on-going cost?
- For the capsule endoscopy scenario, it is mentioned that one less junior doctor is needed for the procedure. But could this be an opportunity cost? In that that doctor has less of a chance to learn? In other words, will something be lost from this scenario from the perspective of deskilling?
Author Response
Dear Reviewer 1,
Thank you very much for your valuable comments. In the attached file you can find a point by point reply to your comments.
Our best regards,
The author team

Reviewer 2 Report
Comments and Suggestions for Authors
Dear Authors,
I have reviewed your manuscript with great interest. Your work addresses the important topic of economic evaluation of AI diagnostic systems, which is highly relevant for healthcare decision-makers. The Cost-Consequence Analysis approach is appropriate for this complex intervention with multiple outcomes. However, several methodological and presentation issues need addressing to strengthen the manuscript. Below are my detailed comments:
- Cost Estimation Transparency: The cost estimates appear oversimplified and lack transparency. Personnel costs are uniformly €5,000 across all technologies without justification, and infrastructure costs vary arbitrarily (€170-€300) between scenarios. Development costs seem underestimated for complex AI systems. Please provide a detailed breakdown of how personnel costs were calculated, including actual time-and-motion studies for staff involvement. Consider differential costs based on technology complexity and, if possible, validate cost estimates through comparison with published literature.
- Sensitivity Analysis: The complete absence of sensitivity analysis undermines the reliability of findings. This is a critical omission for any economic evaluation. Please conduct one-way sensitivity analyses on key cost drivers and, if feasible, perform probabilistic sensitivity analysis using Monte Carlo methods. Test scenarios with different utilization rates and technology adoption speeds to demonstrate the robustness of your conclusions.
- Time Horizon: Although development in this domain is rapid, the one-year analysis may miss important long-term effects of AI implementation. Consider extending the analysis to 3-5 years to capture learning curves, technology maturation, and refresh costs. Please clarify whether this represents a steady-state analysis or implementation year, and discuss long-term clinical outcomes and their economic implications.
- Outcome Validation: Some outcome measures lack clinical validation and meaningful interpretation. Please provide clinical context for statistical differences (e.g., is a 0.08 improvement in clinical performance clinically meaningful?) and define minimal clinically important differences where possible.
- Comparator Definition: "Current practice" is poorly defined across scenarios. Please clearly describe standard-of-care protocols for each scenario, justify why these specific comparators were chosen, and address how current practice varies across different healthcare settings.
- Statistical Reporting: The lack of statistical testing and confidence intervals limits the interpretation of results. Please report p-values for key differences between interventions and comparators, include 95% confidence intervals for all estimates, and describe the statistical methods used for analysis.
- Obstetrics Scenario: Please clarify how the 3-4 week delay in delivery was estimated and validate the conservative nature of the €155,794 savings calculation.
- Echocardiography Scenario: Please justify why increased costs (€7,292) are acceptable for modest accuracy improvements and quantify the clinical value of time savings.
- CCTA Scenario: Please validate the assumption about 159 avoided unnecessary exams and provide more detail on how AI identifies patients who don't need CCTA.
- Literature Integration: Please expand the comparison with existing AI health economics studies and, if available, include discussion of failed AI implementations and their associated costs.
- Decision-Maker Guidance: Consider providing clearer recommendations for different stakeholder groups, addressing varying healthcare system contexts (public vs private), and discussing implications for health technology assessment and phased implementation strategies.
- Limitations: The current limitations section is insufficient. Please include discussion of: the single-center/limited setting nature of studies; proof-of-concept stage limiting generalizability; short time horizon missing long-term effects; lack of patient perspective in outcome measurement; and potential selection bias in participating physicians.
Despite these concerns, your research addresses an important gap in the literature. With the suggested revisions, this manuscript has the potential to make a valuable contribution to the field of AI health economics and inform evidence-based healthcare technology adoption decisions.
I look forward to reviewing the revised version.
Sincerely
Author Response
Dear Reviewer 2,
Thank you very much for your valuable comments. In the attached file you can find a point by point reply to your comments.k
Our best regards,
The author team

Round 2
Reviewer 2 Report
Comments and Suggestions for Authors
Dear Authors,
I have reviewed your revised manuscript with great interest. You have addressed prior concerns adequately. Congratulations.
Sincerely